# Chl1, an ATP-Dependent DNA Helicase, Inhibits DNA:RNA Hybrids Formation at DSB Sites to Maintain Genome Stability in *S. pombe*

**DOI:** 10.3390/ijms23126631

**Published:** 2022-06-14

**Authors:** Deyun He, Zhen Du, Huiling Xu, Xiaoming Bao

**Affiliations:** 1State Key Laboratory of Biobased Material and Green Papermaking, School of Bioengineering, Qilu University of Technology, Jinan 250353, China; duzhen0206@163.com (Z.D.); xuhuiling0906@163.com (H.X.); 2Key Laboratory of Shandong Microbial Engineering, Shandong Academy of Sciences, Qilu University of Technology, Jinan 250353, China

**Keywords:** Chl1 helicase, DNA:RNA hybrid, Rpc37 of RNA Pol III

## Abstract

As an ATP-dependent DNA helicase, human ChlR1/DDX11 (Chl1 in yeast) can unwind both DNA:RNA and DNA:DNA substrates in vitro. Studies have demonstrated that ChlR1 plays a vital role in preserving genome stability by participating in DNA repair and sister chromatid cohesion, whereas the ways in which the biochemical features of ChlR1 function in DNA metabolism are not well understood. Here, we illustrate that Chl1 localizes to double-strand DNA break (DSB) sites and restrains DNA:RNA hybrid accumulation at these loci. Mutation of Chl1 strongly impairs DSB repair capacity by homologous recombination (HR) and nonhomologous end-joining (NHEJ) pathways, and deleting RNase H further reduces DNA repair efficiency, which indicates that the enzymatic activities of Chl1 are needed in *Schizosaccharomyces pombe*. In addition, we found that the Rpc37 subunit of RNA polymerase III (RNA Pol III) interacts directly with Chl1 and that deletion of Chl1 has no influence on the localization of Rpc37 at DSB site, implying the role of Rpc37 in the recruitment of Chl1 to this site.

## 1. Introduction

Living organisms face tremendous challenges which derive either from unavoidable errors during normal DNA replication or from assaults by endogenous or environmental genotoxic agents that cause DNA lesions. Among these lesions, double-strand DNA breaks (DSBs) are highly toxic because failure to repair them leads to chromosome rearrangements and the loss of genetic information [1,2]. Homologous recombination (HR) and nonhomologous end-joining (NHEJ) are two major repair pathways for DSB [3,4]. A critical step in HR involves the nucleolytic processing of one DNA strand from the broken DSB ends to generate long 3′-ended single-strand DNA (ssDNA) overhangs, which is called “end resection” [5]. Nucleases, including the CtIP-MRN (RAD50, MRE11, and NBS1) complex, DNA2 endonuclease, and EXO1 exonuclease, play vital roles in end resection [6,7,8]. Rad51 recombinase protein comes to coat these overhangs with help from Rad52, a central mediator in the HR pathway [9,10]. The Rad51–DNA filament searches for homologous sequences as a template, leading to DNA break repair in an error-free way [11,12,13]. However, the broken DNA ends are simply relegated with little or no procession in the NHEJ pathway, leading to mutations at the break-rejoining sites [14,15]. Thus, the process of DSB ends directly influences genome stability.

Recent studies have shown the discovery of DNA:RNA hybrids at DSB sites [16]. Furthermore, this significantly influences DSB repair pathway choice and/or efficiency [17,18,19]. For example, RNA polymerase II (RNA Pol II) and RNA polymerase III (RNA Pol III) could localize to DSB sites and catalyze DNA:RNA hybrid formation; decreased DNA:RNA hybrid accumulation leads to the suppression of HR efficiency and genetic loss [17,18,19,20]. Moreover, Sen1 helicase could inhibit the illegitimate rejoining between the two ends of the DSB by degrading DNA:RNA hybrids at this locus [21]. The report also showed that, in RNase H-defective strains of *Schizosaccharomyces pombe*, DNA:RNA hybrids accumulated at the resected DSBs and severely impaired the rate of DSB repair [17,22]. Additionally, studies of the miRNA biogenesis enzyme Drosha demonstrated that Drosha is required to control DNA:RNA levels around DSB sites, and mutations in Drosha reduce the repair efficiency of damaged DNA by both the HR and NHEJ pathways [23].

Studies have reported that human ChlR1 helicase can unwind DNA:RNA substrates in vitro [24]. However, whether or not Chl1 regulates the level of DNA:RNA hybrids at DSB and then impacts DNA repair fidelity is still unknown. In vivo assays demonstrate that Chl1 plays multiple roles in cellular DNA metabolism, as is necessary for preserving genome integrity when cells suffer endogenous or exogenous DNA damage [23]. Studies in fission yeast have shown that Chl1 form nuclear foci at HO-induced DSBs [25]. Moreover, ChlR1-depleted HeLa cells have defects in the efficient repair of DSB-induced by bleomycin and endonuclease [26]. Furthermore, research demonstrates that chicken DDX11 acts in concert with the 9-1-1 checkpoint clamp and its loader, Rad17, to promote the repair of bulky DNA lesions by HR [27]. In addition, Chl1 plays a crucial role in sister chromatid cohesion establishment [28,29]. A recent study showed the recruitment of Chl1 to the replisome during replication, which is indispensable for sister chromatid cohesion [26].

Here, we found that Chl1 inhibits DNA:RNA hybrid accumulation at HO-induced DSBs in *S. pombe*. In the *chl1Δ* strains, HR and NHEJ are compromised significantly around the cleavage site, and mutation of RNase H further reduces DNA repair efficiency via these two methods, suggesting that Chl1 promotes DNA damage repair (DDR) by hampering the accumulation of DNA:RNA hybrids at DSB sites. Moreover, we showed that Chl1 physically interacts with Rpc37, an RNA Pol III-specific subunit, suggesting a new role for Rpc5, which participates in DDR through interaction with Chl1.

## 2. Results

### 2.1. Localization of Chl1 at DSB

To investigate the function of Chl1 at DSB sites, we first examined the binding capacity of Chl1-Flags around DSB sites with Flag-beads. *S. pombe* strains harboring an HO endonuclease-induced DNA break system were used [25]. Cells containing this system were shifted from an HO endonuclease gene repression (+ thiamine) condition to an induction (- thiamine) condition in liquid medium to induce HO endonuclease expression, and then HO endonuclease recognized the sequence inserted at the *arg3*^+^ locus to induce DSB. The chromatin immunoprecipitation (ChIP) results demonstrated that the distribution of Chl1 at DSB sites increased with an extended induction time from 16 h to 22 h compared to the preinduction level (Figure 1A), while the HO cut efficiency was also promoted (Figure 1B). In contrast, after 12 h of induction, the HO cut efficiency was close to zero, and the enrichment of Chl1 was similar to preinduction level. The results indicated that Chl1 is an essential factor in DDR.

### 2.2. Rpc37 Recruits Chl1 to DSB Sites

In budding yeast, Chl1 physically associates with the “cohesion establishment factor” Eco1 and plays an important role in sister chromatid adhesion [29,30]. In addition, the interaction between Chl1 and Ctf4 is also required for Chl1′s role in sister chromatid cohesion [31]. Through yeast two-hybrid experiments, we did not find an interaction of Chl1 with Mcl1 (Ctf4 homology) or Eso1 (Eco1 homology) (Appendix A). To test whether Chl1 functions by interacting with other proteins, a Chl1-Flag driven by its natural promoter was constructed. Proteins bound to the Chl1-Flag were affinity purified with Flag beads, and then the Chl1-Flag complex and material derived from the untagged strain were analyzed by SDS-PAGE and Western blot (Appendix A), which implied a number of interacting proteins. Tandem mass spectrometry analysis of purified protein complexes demonstrated that Chl1 interacts with the subunit Rpc37 of RNA Pol III, rather than the untagged comparison (Appendix A). To confirm the interaction between Chl1 and Rpc37 in vivo, we generated strains expressing Chl1-Flag and Rpc37-Myc at their native chromosomal loci. Immunoprecipitation of Chl1-Flag captured endogenous Rpc37 from cell extracts (Figure 2B). Yeast two-hybrid (Y2H) analysis was also used to identify the interaction between Chl1 and Rpc37. The Chl1 and Rpc37 subunits were fused to the GAL4 binding (BD) and activation (AD) domains, respectively. Compared to the control strains, yeast expressing both BD-Chl1 and AD-Rpc37 grew in the absence of adenine and histidine, demonstrating a direct interaction between the Chl1 and Rpc37 subunit (Figure 2A). Since RNA Pol III is known to be localized at DSB sites in U2OS cells [20], we monitored Rpc37 enrichment around the HO cleavage site by ChIP assay in *S. pombe*. As shown in Figure 2C, Rpc37 was recruited to the HO-induced DSB sites, and the deletion of Chl1 did not affect the binding of Rpc37 at this locus, suggesting that Rpc37 recruits Chl1 to DSB sites.

### 2.3. Chl1 Suppresses DNA:RNA Hybrid Accumulation at DSB Sites

Previous research showed that ChlR1, the human homolog of the yeast Chl1, can unwind DNA:RNA hybrids in vitro [24], which implies that Chl1 regulates the levels of DNA:RNA hybrids at DSB sites.

The S9.6 antibody can specifically recognize DNA:RNA hybrids of different lengths. To detect the levels of DNA:RNA hybrids at DSB sites, we immunoprecipitated DNA:RNA hybrids with S9.6, followed by qPCR. As expected, after 19 and 22 h of induction, the *chl1Δ* strain showed strongly increased enrichment of DNA:RNA hybrids compared with the wild type around the cleavage site (Figure 3A,B). Meanwhile, we also found that, with the extended induction time, the distribution of DNA:RNA duplexes at DSB sites increased, which is consistent with the spreading of Chl1 at DSB sites. As shown in Figure 3C, the qPCR signal of hybrids was highly sensitive to treatment with RNase H, further validating the specificity of the antibody. These results suggest that Chl1 inhibits the formation of DNA:RNA hybrids by unwinding DNA:RNA duplexes at the HO cleavage sites.

### 2.4. Chl1 Mutants Display Reduced DNA Repair Efficiency by HR and NHEJ Pathways

To prevent genomic instability or cell death resulting from unrepaired DSB, cells employ HR or NHEJ pathways to repair DNA damage. Here, we investigated whether the mutation of Chl1 suppresses HR-mediated DSB repair by promoting the formation of DNA:RNA hybrids at DSB sites. A site-specific DSB assay was used to quantitate the ratio of marker gene loss, and thus repair responses [32,33]. In this system, an HO endonuclease recognition locus, *MATα,* was integrated into the right arm of a nonessential minichromosome (Ch^16^) (Figure 4A). A strain carrying HO endonuclease gene on genome was induced by the removal of thiamine to generate DSB sites on Ch^16^. Since Ch^16^ is homologous to the centromeric region of chromosome III (Chr III), HR can occur between the two chromosomes, and the loss of marker genes determines the manner of DDR. As presented in Figure 4B, the percentage of marker gene loss was close to zero before HO endonuclease induction. Following DSB induction, Chl1 deletion resulted in a decreased ratio of gene conversion (GC) (62.8%) compared with the wild type (68.3%) (Figure 4C). DNA:RNA hybrids can be degraded by RNase H1 and RNase H201 in *S. pombe* strains [26]. We found that, in the absence of RNase H genes, the GC ratio was further reduced in a *chl1Δ* background, which suggested that the elevated level of DNA:RNA structures in *chl1Δ* mutants suppresses HR efficiency.

Rad52 is a crucial mediator protein during DNA recombination. Our study demonstrated that, when HO endonuclease was induced for 22 h, Chl1 deletion markedly decreased the enrichment of Rad52 at positions 0.2, 2, and 9 kb away from the DSB sites (Figure 4D), indicating that DNA:RNA hybrids reduced HR efficiency by suppressing Rad52 enrichment around HO cleavage sites.

As shown previously, sister chromatid cohesion is crucial for postreplicative DSB repair, and the prevention of cohesin recruitment to DSB sites greatly diminishes repair efficiency [34,35,36]. Rad21 is a conserved structural component of the cohesin complex. In the absence of Chl1, Rad21 accumulation was dramatically reduced around HO-induced DSBs (Figure 4E), and the deletion of both RNase H1 and RNase H201 in *chl1Δ* strain further reduced Rad21 enrichment. This indicated a role for DNA:RNA hybrids in preventing HR by inhibiting Rad21 accumulation at DSB sites.

The procession of the damaged DNA end is a vital determinant of HR or NHEJ repair pathway choice [37,38]. A plasmid-rejoining assay was used to assess the influence of Chl1 deletion on the NHEJ pathway, in which *LEU2* plasmids, linearized with EcoRI (5′overhang), PstI (3′overhang), or PvuII (blunt), were recircularized by NHEJ *in vivo*, and the number of leu+ colonies on plates reflected NHEJ efficiency [33]. In this research, linearized *LEU2* plasmids were transformed into wild-type *chl1Δ* and *chl1Δrnh1ΔΔ* strains, and the leu+ colony number was quantified. The plasmid rejoining ratio in the *chl1Δ* strain was markedly lower than that in the wild type, and deletion of both RNase H1 and RNase H201 in the *chl1Δ* strain further reduced NHEJ efficiency (Figure 5). These findings demonstrated that the increased level of DNA:RNA hybrids in a *chl1Δ* background suppresses HR and NHEJ *in vivo*.

## 3. Discussion

Our study demonstrated that Chl1 helicase is located at DSB sites and that the distribution of Chl1 at HO cleavage sites increased with an extended induction time. Further research found that Chl1 decreased the formation of DNA:RNA duplexes around DSB sites, implying Chl1′s role in unwinding DNA:RNA hybrids. Mutation of Chl1 can significantly decrease the efficiency of HR and NHEJ at the DSB site, and the absence of RNase H enzymes further inhibited DSB repair efficiency at this locus, indicating that Chl1 elevated DNA repair efficiency by suppressing the formation of DNA:RNA structures at the DSB site. In addition, we found that Chl1 interacts directly with the Rpc37 subunit of RNA Pol III, and Chl1 deletion has no influence on the location of Rpc37 around the DSB site, implying the role of Rpc37 in the recruitment of Chl1 to this location.

RNA polymerases I, II, and III are three RNA polymerases which share some subunits in eukaryotes [39]. In fission yeast, RNA Pol II is recruited to DSB sites, whereas a study of U2OS cells demonstrated that RNA Pol III localized to DSB sites and catalyzed RNA synthesis instead of RNA polymerase II [20]. We found the recruitment of Rpc37 at HO cleavage sites; nevertheless, whether RNA pol III is responsible for the formation of DNA:RNA hybrids at this location remains to be studied.

Chl1 is a superfamily 2 (SF2) helicase with ATP-dependent DNA helicase activities [40,41]. The yeast *CHL1* gene is a homolog of two human genes, *ChlR1* and *ChlR2*. The ChlR1 protein was shown to preferentially unwind forked duplex structures with noncomplementary 3′ and 5′ single-stranded DNA and duplexes with a 5′ ssDNA tail, which is consistent with its specific helicase activity on the D-loop structure with a 5′ tail [42,43]. The study also demonstrated that ChlR1 has no helicase activity on Holliday junctions (HJs), although D-loops and Holliday junctions are structures similar to intermediates formed at the early and late stages of HR, respectively. The process of HR-mediated DSB repair consists of DNA end resection, single-strand DNA invasion homologous sequence, and resolution of HJs. The role of Chl1 in end resection and helicase activities on the D-loop structure demonstrated the critical role of Chl1 in preserving genome integrity. It has been reported that Chl1 fails to unwind duplexes that have only a 3′ tail and blunt ends [24,42]; nevertheless, we found that Chl1 can unwind DNA:RNA hybrid substrates with a 3′ tail (unpublished results). Studies have shown that Chl1 moves in the 5′ to 3′ direction on single-stranded DNA, implying that Chl1 recognizes DNA:DNA and DNA:RNA substrates in two distinct manners. Sister chromatids tend to form G-quadruplex (G4) structures during DNA replication or HR, and Chl1 can resolve antiparallel and bimolecular G4 structures with two 5′-tails [44,45], which is consistent with abnormal sister chromatid segregation in the absence of Chl1. Compared with the G4 structure, triplex DNA is a preferred substrate of ChlR1, and it can defend genome integrity by melting the DNA triple helix *in vivo* [46]. Substrate diversity suggests that Chl1 may influence DDR efficiency by simultaneously regulating the levels of multiple DNA intermediates.

A previous publication revealed that Chl1 is critical for sister-chromatid cohesion in budding yeast and mammalian cells [29,47]. Budding yeast *chl1Δ* results in decreased chromosome transmission fidelity or chromosome loss, affecting genome stability. Chromosome loss assays in *S. pombe* showed that the rate of chromosomal loss in the *chl1Δ* mutant was unchanged compared to that in the wild type (unpublished results). In addition, unlike budding yeast [48], the *chl1Δ* mutant was less sensitive to the DNA damage reagent MMS in fission yeast (Appendix A). Studies have shown that both RNase H and Sen1 are recruited to DSB sites and regulate DNA:RNA levels [29,47], suggesting that Chl1, Sen1, and RNase H cooperatively regulate DNA:RNA accumulation and DNA damage repair efficiency at DSB sites.

## 4. Materials and Methods

### 4.1. Yeast Strains and Growth Condition

*S. pombe* strains used in this study are shown in Table 1. Genetic methods used for strain construction were as described [49]. Antibiotic marker genes were cloned from pFA6a-hphMX6, pFA6a-natMX6, and pFA6a-kanMX6 [50]. Cells were cultured on YEA medium (3% glucose, 0.5% yeast extract, and 100 μg/mL adenine) or Edinburgh minimal medium (EMM).

### 4.2. HO-Induced DSB

Strains were first cultured in EMM (Edinburgh minimal medium) with thiamine and were then shifted to thiamine-free EMM for 22 h to induce DSB. After inducing DSB, cells were crosslinked by formaldehyde for a ChIP assay. Strains used in this study were cultured at 30 °C. The induction time of the HO gene was 22 h, unless otherwise noted.

### 4.3. Spot Assay

Exponentially growing cells were collected and normalized to OD_600_ = 0.5. Tenfold serial dilutions of cells were spotted onto YEA plates, with or without the indicated concentration of methyl methanesulfonate (MMS). Cells were cultured at 30 °C for 3–4 days.

### 4.4. Yeast Two-Hybrid Analysis

For yeast two-hybrid analysis, the method described by [51] was used. The pGBKT7 vector and pGADT7 vector were inserted into cDNA, and the constructs were cotransformed into the yeast strain AH109. Transformants were selected on SD/–Trp–Leu plates. Activation of the *HIS3* and *ADE2* reporter genes was assessed on SD/–Ade/–His/–Leu/–Trp plates.

### 4.5. ChIP Assays

ChIP analysis was performed as described previously [52]. Exponentially growing cells were fixed with 3% formaldehyde and then lysed in ChIP lysis buffer (140 mM NaCl, 50 mM HEPES-KOH pH 7.5, 1% Triton X-100, 1 mM PMSF, and 1% deoxycholate) with glass beads. DNA fragments were obtained by sonication, and anti-Flag M2 affinity agarose beads (Sigma, Beijing, China) were used for immunoprecipitation. DNA fragment-bound agarose beads were washed 3 times with ChIP lysis buffer (140 mM NaCl, 50 mM HEPES-KOH, pH 7.5, 1% deoxycholate, and 1% Triton X-100) and twice with wash buffer (10 mM Tris/HCl pH 7.5, 1 mM EDTA, 250 mM LiCl, 1% NP40, and 1% sodium deoxycholate). After crosslinking was reversed, immunoprecipitated DNA was digested with RNase A (Thermo) and Proteinase K (Thermo), followed by phenol/chloroform/isoamylol (25:24:1) extraction. DNA was precipitated by ethanol with 3 M sodium acetate and resuspended in TE buffer. The purified DNA was used for qPCR analysis. For ChIP quantification, we used the following formula as described previously [53]: enrichment = 2^ − ((C*i*^_test_
^− C*i*^_act1_^)IP − (C*i*^_test_
^− C*i*^_act1_^)wce))^, where C*i*_act1_ and C*i*_test_ are the effective amplification cycles for the reference and test, respectively, in the input DNA (wce) samples and immunoprecipitated (IP) samples.

### 4.6. DNA:RNA Hybrid Immunoprecipitation (DRIP)

DRIP was performed as described previously with some modifications [54]. Genomic DNA was extracted from the indicated strains and digested with *Hind* III (Neb), *EcoR* I, *BsrG* I, *Xba* I, and *Ssp* I. The digested DNA was purified and immunoprecipitated with S9.6 antibody (Millipore) in DNA binding buffer (10 mM Na_2_HPO_4_, 140 mM NaCl, and 0.05% Triton X-100) at 4 °C for 16 h. Protein A+G magnetic beads (Millipore) were used for recovering immunoprecipitated DNA and then washed 3 times with DNA binding buffer (50 mM Tris-HCl pH 8.0, 10 mM EDTA, pH 8.0, and 0.5% SDS) and twice with TE buffer. The DNA was eluted from magnetic beads (Millipore) with elution buffer (50 mM Tris-HCl pH 8.0, 10 mM EDTA, pH 8.0, and 0.5% SDS) and then treated with Proteinase K at 50 °C for 3 h. DNA was purified by phenol/chloroform/isoamyl extraction and then used for qPCR.

### 4.7. Coimmunoprecipitation (Co-IP)

Co-IP was performed as described previously [55]. Cells were lysed and then resuspended in 1× HC buffer (150 mM HEPES-KOH pH 7.5, 1 mM EDTA, 1 mM DTT, 250 mM KCl, 10% glycerol, 1 mM PMSF, and 1× Roche protease inhibitor). The supernatants were collected, followed by incubation with anti-Flag M2 affinity agarose beads for 3 h. Agarose beads were washed 3 times with 1× HC buffer and 3 times with 1× PBS buffer. The protein complex was boiled in 1× SDS buffer for Western blotting.

### 4.8. Mass Spectrometry

Exponentially growing cells were lysed and dissolved in 1× HC buffer, Chl1-Flag complex was affinity purified from whole-cell extracts by Flag resin, and then washed 3 times with 1× HC buffer and 1× PBS buffer, respectively. The protein complex bound to agarose beads was eluted with 100 mg/mL 3× Flag peptide (Sigma) twice. The eluted material was concentrated to approximately 100 μL by vacuum centrifugation. A quantity of 10 μL concentrated solution was subjected to silver staining and Western blot, respectively, and the remaining solution was used for liquid chromatography tandem mass spectrometry analysis.

### 4.9. Western Blot

Exponentially growing cells were lysed with glass beads, and the supernatant was collected by centrifugation. Then, an equal volume of 2× SDS buffer was added, followed by boiling for 5 min at 100 °C. Anti-Flag antibody (Sigma) and anti-MYC antibody (CST) were used for Western blot analyses.

### 4.10. Site-Specific DSB Assay

The plasmid rejoining experiment was performed as described previously [33]. A strain carrying the inducible HO endonuclease gene and a background strain without the HO endonuclease gene were cultured in liquid EMM, with or without thiamine, for 48 h, and then they were plated onto amino acid-deficient EMM with agar. The percentage of gene conversion (arg^+^ Hyg^S^ ade^+^ his^+^), NHEJ/SCC (arg^+^ Hyg^R^ ade^+^ his^+^), LOH (arg^+^ Hyg^S^ ade^−^ his^−^), or minichromosome loss (arg^−^ Hyg^S^ ade^−^ his^−^) colonies, with or without HO induction, were calculated by A/B. A and B represent the number of clones with or without HO endonuclease gene, respectively. More than 1000 colonies were counted for each strain, and each experiment was performed 3 times using 3 independently derived strains for all mutants tested.

### 4.11. Plasmid Rejoining Assay

The plasmid rejoining experiment was performed as described previously [56]. In brief, the *LEU* plasmid PS was cut by *Pst* I and *Pvu* II. The *LEU* plasmid PI was cut by *EcoR* I. The excision by restriction enzymes resulted in two identical ends on each of the two plasmids, and the linearized plasmids were transformed into cells. As the plasmids carried a *LEU2* marker, NHEJ frequency was calculated as the percentage of cells grown on leucine-deficient medium over cells transformed with undigested plasmids.

## Figures and Tables

**Figure 1 ijms-23-06631-f001:**
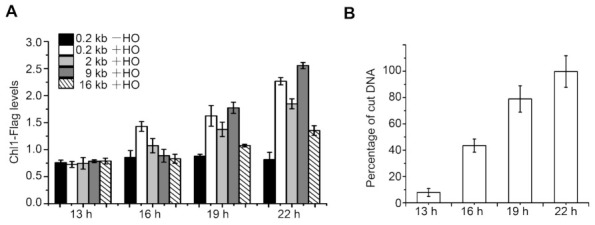
Chl1 is located at DSB sites. (**A**) ChIP assays were performed on the indicated strains before (−HO) and after (+HO) 13, 16, 19, and 22 h of HO endonuclease induction. The quantitative PCR (qPCR) results displayed Chl1 levels at the 0.2, 2, 9, and 16 kb sites from the HO cleavage sites at the indicated induction time. (**B**) ChIP–qPCR results displayed HO cut efficiency after 13, 16, 19, and 22 h of HO induction, qPCR primer used was located on both sides of the DSB. The HO cut efficiency was similar in the HO-induced strains used. Data are displayed as the mean ± SEM of three independent experiments.

**Figure 2 ijms-23-06631-f002:**
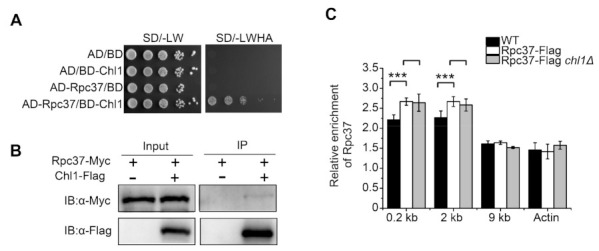
Rpc37 recruits Chl1 to DSB sites by directly interacting with them: (**A**) Gal4 DNA activation (GAL4-AD) and Gal4 DNA binding (GAL4-BD) domains were fused to Rpc37 and Chl1, respectively. The indicated plasmids were co-transformed into AH109 yeast strains; colonies on SD/-Leu-Trp media suggested successful transformation. Cells growing on SD/-Leu-Trp-His-Ade media implied reporter gene expression and an interaction between the two proteins; (**B**) Rpc37-Myc was expressed in cells, either alone or together, with Chl1-Flag, and the Chl1-Flag was immunoprecipitated with anti-Flag attached to agarose beads. The tagged proteins were detected by Western blotting; (**C**) ChIP analyses showed the level of Rpc37 at 0.2 kb, 2 kb, and 9 kb from the HO-induced DSB sites in the indicated strains. The data are presented as mean ± SEM, and three independent assays were performed. Statistical significance was evaluated using an unpaired *t*-test. ***, *p* < 0.001.

**Figure 3 ijms-23-06631-f003:**
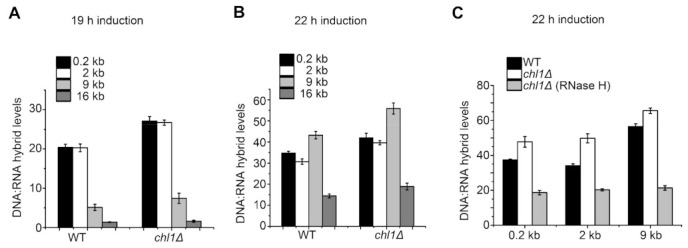
Chl1 reduces DNA:RNA hybrid levels around DSB sites: (**A**,**B**) After 19 and 22 h of HO induction, ChIP–qPCR was performed using antibody S9.6, which specifically recognizes DNA:RNA duplexes in the indicated strains; (**C**) ChIP–qPCR results for the tested strains after 22 h of HO induction, with or without RNase H digestion. The results are represented as the mean ± SEM of three independent experiments.

**Figure 4 ijms-23-06631-f004:**
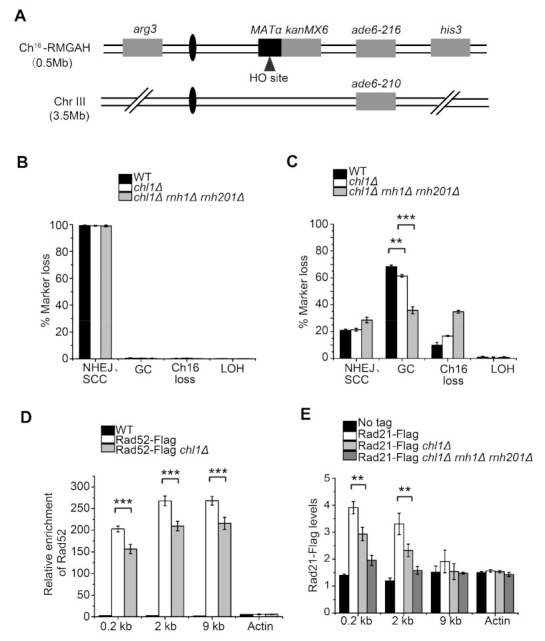
Chl1 influences DNA repair efficiency by HR pathway: (**A**) Schematic of minichromosome Ch16-RMGAH. The *MATα* on the right arm of *Ch^16^* containing an HO endonuclease recognition site is adjacent to *kanMX6*, a G418 resistance marker gene, together with a centromere-distal *his3* gene. The *ade6-M216* heteroallele on Ch^16^ is complemented by *ade6-M210* on Chr III. An *arg3* marker is on the left arm of *Ch^16^*. Derepression of HO endonuclease gene by removal of thiamine generates DSB at the *MATα* recognition site; (**B**,**C**) The percentage of cells with marker gene loss before and after HO endonuclease induction was calculated. The levels of GC, NHEJ/SCC, Ch16 loss, and LOH are presented; (**D**) ChIP–qPCR demonstrated that Chl1 deletion reduced Rad52 accumulation at locations 0.2, 2, and 9 kb from the HO cleavage site with the induction of HO endonuclease for 22 h; (**E**) ChIP–qPCR demonstrated Rad21 levels in the indicated strains at locations 0.2, 2, and 9 kb from the HO cleavage site after 22 h induction of HO endonuclease. Mean ± SEM of 3 replicates is shown. ***, *p* < 0.001; **, *p* < 0.01, Student’s *t*-test.

**Figure 5 ijms-23-06631-f005:**
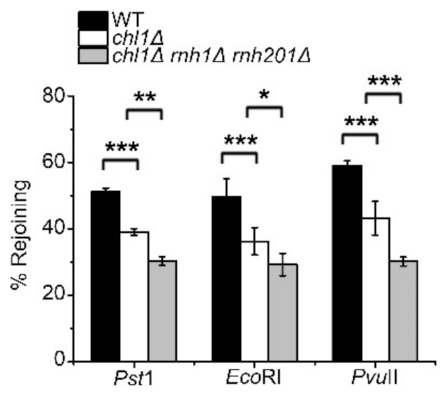
Chl1 is required for NHEJ. The *LEU2* plasmids (pAL19) were linearized with *Pvu* II, *Pst* I, and *EcoR* I. The linearized plasmids were transformed into wild-type *chl1Δ,* and *chl1Δ rnh1Δ rnh201Δ* strains; the percentage of cells grown on leucine-deficient medium represented plasmid rejoining efficiency. Mean ± SEM of 3 replicates is shown. ***, *p* < 0.001; **, *p* < 0.01; *, *p* < 0.05, Student’s *t*-test.

**Table 1 ijms-23-06631-t001:** Strains used in this study.

Strain Name	Genotype
LLD3650	*h- his3D1 ura4-D18 crb2Δ::ura4 rad22-CFP::KanMX6 arg3::HOsite-KanMX4 leu1-32::YFP-Crb2::leu1 nmt41-HO-his3*
ywp746	*h- his3D1 ura4-D18 crb2Δ::ura4 rad22-CFP::KanMX6 arg3::HOsite-KanMX4 leu1-32::YFP-Crb2::leu1 nmt41-HO-his3 chl1-flag::HphMX6*
SPJ1577	*h+ Leu1-32 ura4DS/E*
ywp168 ywp1545	*h+ Leu1-32 ura4DS/E chl1Δ::HphMX6* *h+ Leu1-32 ura4DS/E rnh1∆::NatMX4 rnh201∆::KanMX4 chl1∆::HphMX6*
ywp702	*h+ leu1-32 ura4 DS/E Chl1-5xflag::KanMX6*
ywp1108	*h+ leu1-32 ura4 DS/E rpc37-13xmyc::NatMX6 Chl1-5xflag::HphMX6*
ywp1106	*h+ leu1-32 ura4 DS/E rpc37-13xmyc::NatMX6*
ywp1005	*h- his3D1 ura4-D18 crb2Δ::ura4 rad22-CFP::KanMX6* *nmt41-HO-his3arg3::HOsite-KanMX4 leu1-32::YFP-Crb2::leu1 rpc37-5xflag::HphMX6 chl11Δ::NatMX6*
ywp1328	*h- his3D1 ura4-D18 crb2Δ::ura4 rad22-CFP::KanMX6* *nmt41-HO-his3 arg3::HOsite-KanMX4 leu1-32::YFP-Crb2::leu1* *rpc37-5xflag::HphMX6*
ywp1330	*h- his3D1 ura4-D18 crb2Δ::ura4 rad22-CFP::KanMX6* *nmt41-HO-his3 arg3::HOsite-KanMX4 leu1-32::YFP-Crb2::leu1 chl1Δ::NatMX6*
ywp635 ywp637	*h+ arg3-D4 Ch16-RMGAH leu1::nmt41-Leu2* *h+ arg3-D4 Ch16-RMGAH leu1::nmt41-HO-Leu2*
ywp659 ywp661	*h+ arg3-D4 Ch16-RMGAH leu1::nmt41-Leu2 chl1Δ::HphMX6* *h+ arg3-D4 Ch16-RMGAH leu1::nmt41-HO-Leu2 chl1Δ::HphMX6*
ywp1705	*h+ arg3-D4 Ch16-RMGAH leu1::nmt41-Leu2 chl1Δ::HphMX6* *rnh1Δ::NatMX6 rnh201Δ::KanMX6*
ywp1707	*h+ arg3-D4 Ch16-RMGAH leu1::nmt41-HO-Leu2 chl1Δ::HphMX6* *rnh1Δ::NatMX6 rnh201Δ::KanMX6*
ywp852	*h- his3D1 ura4-D18 crb2∆::ura4 nmt41-HO-his3 arg3::HOsite-KanMX4 leu1-32::YFP-Crb2::leu1 rad52-5xflag::HphMX6*
Ywp992	*h- his3D1 ura4-D18 crb2∆::ura4 nmt41-HO-his3 arg3::HOsite-KanMX4 leu1-32::YFP-Crb2::leu1 rad52-5xflag::HphMX6 chl11∆::NatMX6*
ywp1709	*h- his3D1 ura4-D18 crb2∆::ura4 rad22-CFP::KanMX6 nmt41-HO-his3 arg3::HOsite-KanMX4 leu1-32::YFP-Crb2::leu1 rad21-5xflag::HphMX6 chl1∆::NatMX6 rnh1Δ::NatMX6 rnh201Δ::KanMX6*

## Data Availability

The data presented in this study are available on request from the corresponding author.

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
