# Peer review of "Chl1, an ATP-Dependent DNA Helicase, Inhibits DNA:RNA Hybrids Formation at DSB Sites to Maintain Genome Stability in *S. pombe"

_ijms, 2022, doi:10.3390/ijms23126631_

Round 1
Reviewer 1 Report
He et al describe the activities of the yeast helicase Chl1 and is potential role in genome stability. The main issue of the manuscript is the lack of details about the methods. So, I found it difficult to judge the validity of the conclusions draw by the authors. A few examples:
There is no description of the cells used in the study (S.cerevisae? genotype?...)
Figure legends and plot axis do not show enough information:
Fig1: what are CHl1-flag levels? What is the scale referring to?
Intact cut sites (%): the plot shows that only 1 % are intact in the absence of OH?
What is the purpose of the inset if both bars are in white color?
The helicase seems to be present as far as 9kb from the cut so how specific this would be of DSB?
Figure 2 A: the mass spec experiment requires a full description of what was done and a more descriptive figure showing the results
Fig3: same comment as before, how specific is the presence of DNA/RNA hybrid accumulation at DSB if they are detected 9 kb apart?
In all the plots, it would be clearer and more convincing to show an example of the experiment used for quantification.
Sometimes authors refer to ‘Means plus minus S.E.M’(fig1) , others ‘Means plus minus s.e.’(fig4) or none at all (fig2).
There is no figure legends for the supplementary figures so it is hard to understand them.
Reviewer 2 Report
In their manuscript entitled “Chl1 prevents DNA:RNA hybridization at double-strand breaks to maintain genome stability”, the authors primarily analyze the relationship of DNA helicase, Chl1, to double-strand DNA breaks (DSBs), particularly its accumulation at the DSB locations. They used mating type switching-specific yeast endonuclease to create DSBs and looked for Chl1 using CHIP assays. Also, they investigated the role of RNA pol III associated RPC37/RPC5 subcomplex in this process. Overall, this study is novel and addresses specific role of Chl1 in DNA repair pathway, however, there are several weaknesses that need to be addressed before the manuscript is accepted for publication. Following are the major points that authors should address:
Abstract:
- There are a few typographical errors (for example: todouble-strand) that should be fixed. Also in the last sentence, authors use the acronyms RPC37and RPC5 in relation to Chl1, although they are part of the same subcomplex. Why not add an introductory sentence to introduce the rational for investing this subcomplex so that readers who are not from the same field of study do not get confused?
- ‘Sister cohesion’ of sister chromatid cohesion?
Introduction:
- Authors should discuss what are Rpc37 / Rpc5 in relation to RNA Pol III and why they wanted to look into association of these factors with Chl5, instead of randomly bringing up the results they have obtained.
- What is ‘HO’ or ‘HO endonuclease’? Nowhere in the manuscript authors mentioned the rationale of using this yeast enzyme for creating DSBs. This should be clarified.
Results:
- Regarding localization of Chl1 at DSBs, authors should elaborate on the experimental designs with proper logical explanations. Currently, the results are described in a manner that provides no clear and precise understandings on why and how these experiments were done. “Strains with an engineered HO endonuclease….” - which strains? why P41 nmt1 is important here? Which antibody used for CHIP assay? What are the controls used in this assay that provides specificity of Chl1 binding to DSBs? How do they determine the exact positions of the DSB sites? Can they show the gel image from RT-PCR (or cite if they show in the supplement?)
- While describing the results on observation that Rpc37 recruits Chl1 to DSBs, authors report identifying Rpc37 interaction with Chl1 by mass spectrometry using chl1-Flag as a bait. However, authors provide no information about how this mass spectrometry was done and what are the controls used in this set of experiments. How many replicates were used and what were the spectral counts of Rpc37 in each of the replicate? What was the spectral count for the bait protein? If Chl1 recruitment to DSBs is dependent on Rpc37, does Rpc37 knockdown reduce localization. of Chl1 to DSBs? Author should provide this important validation.
- Regarding the observation that Chl1 suppresses DNA/RNA hybrid accumulation at DSBs, the graph shown in Figure 3 is confusing. What are the values shown in on the y-axis? are they absolute Ct values or some kind of log values? This experiment does not convincingly show that Chl1 reduces DNA/RNA hybrid levels around DSBs. The standard error shown in 0.2kb data point from Chl1 depleted cells is merely similar to what. is observed in WT. What are the values obtained from a control IP and how the data was normalized?
4) Similar problems are evident in experiments described in Figure 4. Authors did not elaborately describe their rational, hypothesis or experimental conditions. For example, Figure 4A and B shows the percentage of cells with marker loss before and after DSB induction. However, how this data was calculated is not clearly described.
5) Materials and methods sections missing detailed description of the experiments and in some cases, the resources of the reagents/strains used in the experiments.
Round 2
Reviewer 1 Report
The authors have addressed the requested points and have improved considerably the quality of the manuscript.